# Effect of TB Treatment on Neutrophil-Derived Soluble Inflammatory Mediators in TB Patients with and without HIV Coinfection

**DOI:** 10.3390/pathogens12060794

**Published:** 2023-06-01

**Authors:** Nádia Sitoe, Imelda Chelene, Sofia Ligeiro, Celso Castiano, Mohamed I. M. Ahmed, Kathrin Held, Pedroso Nhassengo, Celso Khosa, Raquel Matavele-Chissumba, Michael Hoelscher, Andrea Rachow, Christof Geldmacher

**Affiliations:** 1Instituto Nacional de Saúde, Marracuene 3943, Mozambiquecelso.khosa@ins.gov.mz (C.K.);; 2CIH LMU Center for International Health, Ludwig-Maximilians University, 80802 Munich, Germany; 3Division of Infectious Diseases and Tropical Medicine, University Hospital, LMU Munich, 80802 Munich, Germany; mo.i.mahmoud@gmail.com (M.I.M.A.); geldmacher@lrz.uni-muenchen.de (C.G.); 4German Center for Infection Research, Partner Site Munich, 80802 Munich, Germany; 5Fraunhofer Institute for Translational Medicine and Pharmacology ITMP, Immunology, Infection and Pandemic Research, 80799 Munich, Germany

**Keywords:** patients living with HIV, TB, multiplex, TB treatment

## Abstract

The mycobacteriological analysis of sputum samples is the gold standard for tuberculosis diagnosis and treatment monitoring. However, sputum production can be challenging after the initiation of TB treatment. As a possible alternative, we therefore investigated the dynamics of neutrophil-derived soluble inflammatory mediators during TB treatment in relation to HIV ART status and the severity of lung impairment. Plasma samples of TB patients with (N = 47) and without HIV (N = 21) were analyzed at baseline, month 2, month 6 (end of TB treatment) and month 12. Plasma levels of MMP-1, MMP-8, MPO and S100A8 markedly decreased over the course of TB treatment and remained at similar levels thereafter. Post-TB treatment initiation, significantly elevated plasma levels of MMP-8 were detected in TB patients living with HIV, especially if they were not receiving ART treatment at baseline. Our data confirm that the plasma levels of neutrophil-based biomarkers can be used as candidate surrogate markers for TB treatment outcome and HIV-infection influenced MMP-8 and S100A8 levels. Future studies to validate our results and to understand the dynamics of neutrophils-based biomarkers post-TB treatment are needed.

## 1. Introduction

Tuberculosis (TB) and HIV coinfection is a public health problem in TB-endemic countries such as Mozambique [1]. HIV infection increases the risk of developing active TB disease upon infection and of reactivating latent TB [2,3]. The diagnosis of active TB is commonly based on the use of sputum samples [4,5] via the detection of acid fast bacilli using smear microscopy, the culture detection of *Mycobacterium tuberculosis*, or by molecular diagnosis [6,7,8,9]. HIV infection reduces the sensitivity of both the GeneXpert molecular test [10,11] and smear microscopy [6,12] as it is often associated with paucibacillary tuberculosis. Both techniques, sputum smear microscopy and mycobacterial culture, have low sensitivity and modest specificity for predicting TB treatment failure [13]. This can be partly explained by the fact that treatment reduces the ability of patients to produce sputum. Therefore, treatment monitoring using sputum-independent host response biomarkers in blood would be a potential and feasible alternative to the laborious and time-intensive microbiological culturing of sputum.

Various blood-based biomarker signatures aiding in the diagnosis of active TB disease and in monitoring TB treatment [4,5,14,15,16,17] and the severity of pulmonary impairment [18,19,20] have been proposed. The systemic host response before and after TB treatment initiation can probably be used to predict pulmonary and treatment outcomes; however, it is still unknown to what extent this is possible [21]. Macrophage and neutrophil levels and activation markers, inflammatory mediators, matrix metalloproteinase (MMP) levels and other factors involved in the host inflammatory response to TB are linked to the inflammatory response that influences the clinical outcome of TB [21]. Rambaran et al. (2020) identified plasma signatures associated with TB culture conversion 8 weeks after the initiation of TB treatment and with lung cavitation in active TB cases [15]. Muefong and Sutherland (2020) described that levels of pro-inflammatory neutrophil-derived mediators in blood, such as MMP-1, -2, -3, -8 and -9, are linked to lung damage at baseline and lung recovery [18,22]. Consistent with these results, another study also showed that levels of certain neutrophil mediators decreased upon TB treatment initiation in a cohort of TB patients living with HIV; hence, such MMPs can be potential biomarkers for monitoring TB treatment regardless of HIV coinfection [23].

Nonetheless, little is known about the profile of blood-based neutrophil mediators at the end of TB treatment and whether they may have potential to predict the outcome of TB in the context of the severity of lung impairment in high HIV and TB burden settings. Our study therefore aimed to study the concentrations of neutrophil-derived proinflammatory mediators during TB treatment in the context of HIV infection and post-TB lung impairment.

## 2. Material and Methods

### 2.1. Study Population

TB patients with a positive result on culture methods or Xpert MTB/RIF or Ultra (Cepheid, Sunnyvale, CA, USA) were recruited at the Machava General Hospital and Mavalane Health Center, in Maputo, Mozambique, as part of the TB sequel study [24]. This study aims to understand the clinical, microbiological, immunological and socio-economic risk factors affecting the long-term outcomes of pulmonary TB in four African countries. All participants who consented to participate in the study were tested for HIV following the national testing algorithm, and those who knew their HIV status were asked about their ART status. Participants were followed for 24 months after the initiation of TB therapy over nine visits: at baseline (BL), after 14 days, and after 2 (M2), 4, 6 (M6), 9, 12 (M12), 18 and 24 months. Sixty-eight subjects were selected for our analyses based on the availability of plasma samples at BL, M2, M6 and M12, HIV and ART status information, and TB culture results at months 2 and 6. Sixty-five participants received standard TB treatment, an intensive phase for two months and a continuous phase for four months accordingly to the WHO TB treatment guideline [25]. Three study participants were TB-drug-resistant and had a different regimen of TB treatment [26].

For all participants living with HIV (PLHIV), the CD4+ T-cells counts at baseline were determined. During each study visit, liquid and solid TB cultures (Lowenstein-Jensen) were performed. The study protocol was approved by the Ludwig-Maximillians University (approval number 786-16, from 10 December 2018) and Mozambican Ethical Committees (approval number 292/CNBS/21, from 31 May 2021).

### 2.2. Assessment of Lung Function and Damage

Spirometry was carried out at each study visit to assess lung function over the time of TB treatment and thereafter to assess pulmonary function impairment. The American Thoracic Society/European Respiratory Society (ATS/ERS) guidelines were used to interpret the spirometry results. The values for forced ventilatory capacity (FVC) and forced expiratory volume in one second (FEV1), as well as the FEV1/FVC ratio, were standardized for age, sex and height using the Global Lung Function Initiative (GLI) reference equations. The pulmonary function impairment was categorized as mild for an FVC value and FEV1/FVC ratio > 85% of the predicted value, moderate for an FVC value or FEV1/FVC-ratio 55–85% of the predicted value, and severe for an FVC value or FEV1/FVC-ratio < 55% of the predicted value [27]. As severe TB disease was experienced at baseline by some participants, not all baseline spirometry data were conclusive; therefore, spirometry data from the first study visit, day 0 and day 14 were used herein as baseline lung function determinants.

Pulmonary damage was determined by independent readers of thoracic X-ray results and graded using a Ralph scoring system [28].

### 2.3. Luminex Assay

Whole blood was collected in heparin tubes, centrifuged at 2500 rpm for 10 min for plasma separation and stored at −80 °C. Plasma samples tested in different batches were diluted 1:2, and the levels of plasma biomarkers were measured using commercial kits, Human Magnetic Luminex assays (Lot number L129636 R&D, Minneapolis, MN, USA), as previously published [29]. The kit detected 15 soluble proteins, namely, IFN-gamma, NCAM, TNF-alpha, IL-8, IL-10, IL1b, GM-CSF, IL-13, IL-12, CD40-ligand, MMP-1, MMP-2, MMP-8, S100A8 and MPO. The sample plates were read on the same day using a MAGPIX system, xMAP instrument (Luminex, Austin, TX, USA). Finally, xPONENT software, version 4.3 (Luminex, Austin, TX, USA), was used for bead acquisition and analysis.

### 2.4. Statistical Analysis

The continuous clinical and demographic characteristics of people living with and without HIV were compared using the Mann–Whitney non-parametric test.

Protein plasma levels were compared between groups at different time points (baseline and at months 2, 6 and 12) using Kruskal–Wallis and Mann–Whitney non-parametric tests. The Wilcoxon signed rank test was used for comparisons among the study groups at different time points. Spearman’s coefficient was used to assess the correlation of the plasmatic biomarkers with the neutrophil counts and the Ralph score at baseline and months 6 and 12. All statistical analyses were performed in GraphPad Prism software, version 5 (GraphPad software, San Diego, CA, USA). *p* values < 0.05 were considered statistically significant.

## 3. Results

### 3.1. Characteristics of Study Population

Table 1 provides an overview of patient characteristics stratified by HIV status. Sixty-eight TB sequel study participants with a median age of 38.2 years [min–max: 19.01–60.60] were selected for the analyses of plasma protein markers based on HIV and ART status and on the completeness of the relevant data on lung damage and function after the end of TB treatment. The majority of the participants were male 57.35% (39/68), and 69.12% (47/68) were living with HIV. Among the active TB (aTB) patients living with HIV, 48.94% were ART-naïve at enrollment, and the median CD4 T-cell count was 134.0 cells/mm^3^ [95% CI: 66.50–343.0]. The majority of the subjects then received standard TB treatment of 6 months upon study inclusion (65/68) except for three participants, who had TB caused by rifampicin-drug-resistant MTB and were treated for 8 months instead of 6.

Six months of TB treatment reduced lung damage, as assessed via X-ray, with declining median Ralph scores from 15 at baseline before treatment to 5.00 after 6 months of TB treatment (*p* < 0.0001). TB treatment also reduced the proportion of patients with lung impairment; 75.76% of the patients had a lung impairment at baseline (BL) compared to 61.22% at month 6 and 60.78% at month 12 post TB treatment initiation. Moreover, TB treatment reduced peripheral blood neutrophil levels (median: BL = 3.85 cells/mm^3^ and M6 = 1.61 cells/mm^3^; *p* = 0.01), consistent with a reduction in TB-associated systemic inflammation after treatment completion.

### 3.2. MMP-2 Plasma Levels Strongly Correlate with Neutrophils and Ralph Score, and Increased after TB Treatment Initiation

First, we analyzed the plasma concentrations of the inflammatory biomarkers (MMP-1,-2,-8, S100A8 and MPO) over the course of TB treatment within the whole cohort, regardless of HIV status or clinical outcomes. The concentration of MMP-2 moderately increased during the 6 months of TB treatment (median: 38,290.0 pg/mL at BL versus 40,080.0 pg/mL at M2 (*p* < 0.0001) and 41,450.0 at M6 (*p* = 0.0032)) but did not change between months 6 and 12 (median: 41,450.0 pg/mL versus 42,030.0 pg/mL, *p* = 0.4417) (Figure 1B). In contrast, consistent with the results reported by Muefong et al. (2021) [29], the overall plasma concentrations of MMP-1, MMP-8, S100A8 and MPO were significantly reduced from BL to months 6 and 12. These reductions were most pronounced at 2 months after TB treatment initiation, while the concentrations of these analytes did not change between months 6 and 12 (Figure 1A,C–E).

Neutrophils contribute to the development of lung cavities [18,30], and we therefore assessed the correlation of the neutrophils and the concentrations of the plasma biomarkers MMP-1, MMP-2, MMP-8, MP0 and S100A8 at BL, month 6 (end of TB treatment) and month 12 (6 months post TB treatment). Baseline neutrophil counts were inversely correlated to the levels of MMP-2 at baseline (r = −0.42 and *p* = 0.0096) and month 6 (r = −0.46 and *p* = 0.014), and positively correlated with MMP-1 at month 12 (r = 0.46 and *p* = 0.04) (Table 2). We also found that the neutrophil count at month 12 inversely correlated with the levels of MMP-1 (r = −0.35 and *p* = 0.046) and S100A8 (r = −0.46 and *p* = 0.48 and *p* = 00009) at month 6 (Table 2).

Then, we investigated the correlation of these plasma biomarkers with lung damage (Table 3). The baseline Ralph score did not show any association with BL plasma biomarkers. However, the baseline Ralph score positively correlated with month 6 plasma levels of MMP-8 (r = 0.36, *p* = 0.04) and MPO (r = 0.39, *p* = 0.02) and negatively correlated with MMP-2 (r = −0.37 and *p* = 0.03); the Ralph score at month 6 correlated with the level of MMP-8 at month 12 (r = 0.52 and *p* = 0.02).

We also tested for correlations between plasma biomarkers amongst each other and found that S100A8 correlated to MMP-1 and MMP-8 at baseline and at months 6 and 12. S100A8 and MPO, however, correlated at months 6 and 12 (M6: r = 0.38 (*p* = 0.02) and M12: r = 0.69 (*p* < 0.0001)), respectively. MPO correlated with MMP-2 at baseline and month 12, and it also correlated with MMP-8 at months 6 and 12 (Table 4).

### 3.3. ART-Naïve Patients Living with HIV Coinfected with TB Exhibit Markedly Elevated Levels of MMP-8 and S100A8

Next, we studied the concentrations of MMP-1, 2, 8, S100A8 and MPO analytes in relation to HIV and ART status. HIV infection status did not alter MMP-1 levels over the entire period study (Figure 2A). PLHIV on ART with active TB had significantly elevated MMP-2 plasma concentrations at baseline and at months 6 and 12 compared to TB mono-infected patients (Figure 2B). PLHIV who were ART-naïve had 2.1-fold increased concentration levels of MMP-8 at baseline compared to TB mono-infected patients (median: 10,050.0 pg/mL versus 4,805.0 pg/mL, *p* = 0.03). Similar differences, albeit at a lower level, were observed at months 2 (median: 5,097.0 pg/mL versus 3,893.0 pg/mL, *p* = 0.031) and 6 (median: 3,980.0 pg/mL versus 2,215.0 pg/mL, *p* = 0.018). Furthermore, plasma levels of MMP-8 were also higher in ART-naïve PLHIV compared to individuals on ART at months 2 (*p* = 0.048) and 6 (*p* = 0.045) (Figure 2C). At the end of treatment, we had similar findings in the concentration levels of S100A8 in ART-naïve PLHIV compared to individuals on ART (median: 357.2 pg/mL versus 311.9 pg/mL, *p* = 0.043) (Figure 2D).

Regarding plasma levels of MPO, PLHIV on ART and ART-naïve PLHIV had higher concentrations compared to TB mono-infected patients at baseline (median: 47,230.0 pg/mL versus 37,790.0 pg/mL, *p* = 0.019) and month 6 (median:42,050.0 pg/mL versus 32,060.0 pg/mL, *p* = 0.016), respectively (Figure 2E).

There were three PLHIV with TB-drug resistance at baseline in our study; one of them was ART-naïve at BL. At month 6 of TB treatment, the PLHIV ART-naïve patient developed cavities in the lungs (RS = 15.0 at BL and RS = 60.0 at month 6). In summary, these data show that MMP-2, MMP-8, S100A8 and MPO neutrophile-derived inflammatory markers are elevated in PLHIV before and after the initiation of TB treatment.

### 3.4. Higher Levels of MMP-8 and MPO before Treatment Initiation Are Linked to More Severe Lung Impairment at the End of TB Treatment Initiation

Based on spirometry results at baseline and month 6, we grouped TB sequel participants with normal and mild lung impairment as the “less severe” lung impairment group and those with moderate and severe lung impaired as the “more severe” lung impairment group. Of note, participants not able to undergo a spirometry test at baseline mostly did so on day 14 of the study.

Considering the spirometry results from BL/D14 visits and month 6, the levels of MMP-1 and MMP-2 were similar between the patients with more severe and less severe lung impairment at all study points (Figure 3A,B). However, using the BL/D14 spirometry, patients with less severe lung impairment had higher plasma levels of MMP-8 and S100A8 at the month 12 visit compared to those with more severe lung impairment, *p* = 0.0007 and *p* = 0.018, respectively (Figure 3C,D). Moreover, using the BL/D14 spirometry results, we observed that 6 months after the initiation of TB treatment, plasma levels of MPO in the more severe patients were higher than the less severe patients (median: 40,430.0 pg/mL versus 30,460.0 pg/mL, *p*= 0.0134) (Figure 3E).

Interestingly, before TB treatment initiation, patients with more severe lung impairment after TB treatment at month 6 had higher levels of MMP-8 and MPO compared to those with less severe lung impairment, *p* = 0.016 and *p* = 0.001, respectively. After TB treatment initiation, the levels of these two biomarkers were comparable between the two groups, although at months 2 and 6, MPO tended to be higher in the more severe compared to the less severe lung-impaired patients (Figure 4C,E). The levels of S100A8 were also comparable between subjects with more and less severe lung impairment (Figure 4D).

The Ralph score at baseline was not significantly associated with the Ralph score at the end of TB treatment (data not shown; r= 0.23 and *p* = 0.087), and the patterns of the plasma levels of MMP-8 and MPO in subjects with more severe lung impairment at enrollment were similar to those with any lung impairment at month 6 (Figure 4F).

## 4. Discussion

Our study investigated several neutrophil-derived inflammatory mediators in relation to TB disease severity, HIV status and treatment outcome. Plasma levels of MMP-1, MMP-8, S100A8 and MPO were markedly reduced after TB treatment initiation regardless of HIV infection status, with the most noteworthy decline occurring by month 2. Nonetheless, HIV coinfection influenced the plasma levels of some of these mediators; MMP-8, S100A8 and MPO were elevated at baseline and also at the end of TB treatment, albeit to a lower degree. The notable reduction in these inflammatory mediators at month 2 aligns with sputum conversion; all participants were sputum negative by month 2 post TB treatment initiation. This indicates that changes in the levels of these inflammatory mediators are linked with the early bactericidal effect during the first two months of intensive TB treatment. Similar results were observed in previous studies [18,19,23,29,31,32]. Neutrophil levels in blood correlated negatively with MMP-2 at baseline and month 6, and they also declined during TB treatment, with the opposite effect observed for MMP-2 levels. The other neutrophil-derived mediators did not correlate with neutrophil levels at baseline. One possibility is that it is particularly the activated neutrophils in the infected lung tissue that release these mediators. An in vitro study has shown that neutrophils in direct contact with MTB bacilli express high levels of MMP-8 [33]. Muefong and colleagues reported positive correlations of baseline Ralph scores and neutrophil counts with several neutrophil-derived inflammatory markers, such as MMP-8 and S100A8 [29]. These correlations were often not detected in our study. The abundance of low-density neutrophils in aTB patients is another possibility. These neutrophils are characterized by the inability to phagocyte the MTB and to produce reactive oxygen species [34], both of which are important for release of the neutrophils-derived inflammatory markers. HIV infection was associated with MMP-8 baseline levels in our study, and it is often associated with paucibacillary disease [35,36] and fewer cavities [37,38]. The high proportion of PLHIV and the low numbers of subjects with Ralph scores above 40 (indicative of cavities) and participants with CD4 counts below 200 in our study could therefore be responsible for these discrepancies. Moreover, our study had a smaller statistical power compared to the Muefong study.

It has been reported that HIV infection modulates the expression of matrix metalloproteinases (MMPs) [39]. In our cohort, the plasma levels of MMP-8 were higher in PLHIV, particularly those who were ART-naïve at baseline, even at the end of TB treatment and after ART initiation. Hence, other factors related to HIV infection, such the impact of HIV ART initiation, higher levels of systemic inflammation and/or microbial translocation probably contribute to this observation [40,41,42,43,44]. Our results are consistent with Alisjahbana et al. (2022), who found that a low lymphocyte level was significantly correlated to higher levels of MMP-8 [40]. Furthermore, TB-immune reconstitution inflammatory syndrome (TB-IRIS) often occurs in highly immunosuppressed patients after the initiation of ART [45]. Ravimohan et al. (2016) found an increase in the plasma levels of MMPs, including MMP-8, associated to TB-IRIS and a decrease in lung function post TB cure [41]; however, we did not specifically assess the proportion of PLHIV who developed TB-IRIS in our study.

Plasma levels of MMP-1, MMP-8, S100A8 and MPO before and after TB treatment initiation or 6 months after TB treatment were strongly correlated. This was also observed in animal and human models [46,47]. Gonzalez-Lopez et al. (2012) found that serum and plasma levels of MMP-8 modulate the levels of S100A8/9 [46], which is a contribution of the genetic polymorphisms of S100A8/9 genes [47]. In our study, participants with less severe levels of lung impairment had higher plasma levels of MMP-8 and S100A8 at month 12 post TB treatment initiation. Neutrophil-related inflammatory mediators correlate with pulmonary injury in humans [18,29,48] and animals [30,49,50]. S100A8, MMP-8 and MPO are involved in lung pathology [5,18,29,30,51] as consequences of lung tissue degradation in tuberculosis [18,29,30], emphysema [52], COVID-19 [53] or COPD [54]. Hence, while speculative, subjects starting TB treatment with less severe lung impairment may potentially develop a more severe lung condition after TB treatment as MMP-8 and S100A8 biomarkers are involved in the pulmonary damage. Indeed, at the end of TB treatment or 6 months after TB treatment, the plasma levels of the inflammatory mediators MMP-8, MPO and S100A8 were associated with the Ralph score, consistent with the notion that these biomarkers may contribute to lung damage and the impairment of lung function after the end of TB treatment [55]. However, these findings must be confirmed in future studies with larger sample sizes and long-term follow ups.

The limitations of this study include the relatively small sample size of patients at 6 months post TB treatment and of those with mild and severe lung impairments. In addition, our study was of an exploratory nature, and we did not seek to confirm these results with an independent experiment and samples. Other limitations include the lack of availability of plasma HIV viral load data because it has been reported that HIV drives the dysfunction of neutrophils [56] that could probably influence the production and release of its inflammatory mediators.

In summary, our study confirmed that the plasma levels of neutrophil-based biomarkers markedly change with TB treatment regardless of HIV ART status and the severity of lung impairment. HIV infection status and ART negativity at baseline influenced MMP-8 and S100A8 levels. Future studies to validate our results and to explore the post-TB treatment dynamics of neutrophil-based biomarkers are needed.

## Figures and Tables

**Figure 1 pathogens-12-00794-f001:**
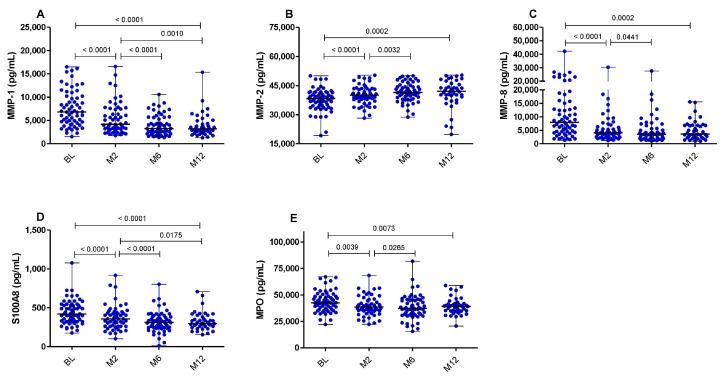
Dynamic changes in the concentrations of neutrophil-related inflammatory biomarkers upon TB treatment initiation. The concentrations of the biomarkers MMP-1 (**A**), MMP-2 (**B**), MMP-8 (**C**), S100A8 (**D**) and MPO (**E**) in all subjects (n = 68) at baseline (n = 62), 2 months (n = 56), 6 months (n = 59) and 12 months (n = 42) after TB treatment initiation are shown. Bars represent medians and interquartile range. Statistical analyses were performed using the Kruskal–Wallis and the Wilcoxon signed rank test for paired samples. A *p*-value of < 0.05 was considered significant.

**Figure 2 pathogens-12-00794-f002:**
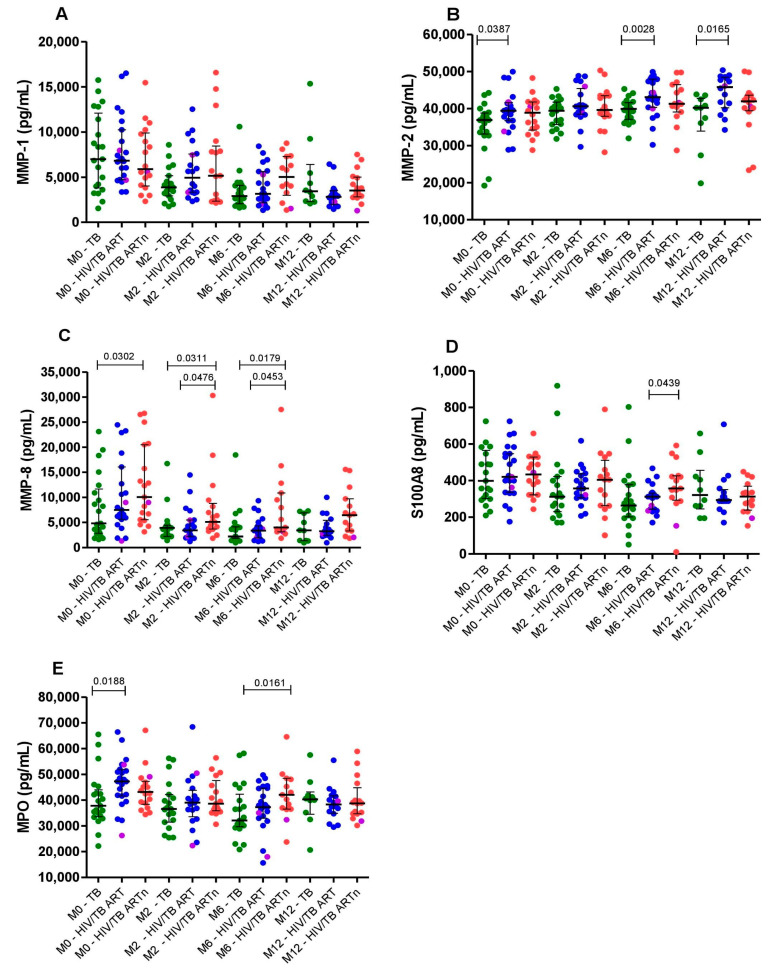
Comparisons of neutrophil-related inflammatory biomarkers MMP-1 (**A**), MMP-2 (**B**), MMP-8 (**C**), S100A8 (**D**) and MPO (**E**) between TB mono-infected (n = 21, green dots), HIV/TB coinfected patients on ART (n = 23, blue dots) and HIV/TB coinfected ART-naïve patients (n = 19, red dots) at baseline, month 2, month 6 and month 12 study visits. The magenta dots are the subjects with drug-resistant TB. Bars represent the median and interquartile range. The Mann–Whitney test was used for comparisons between the groups at different time points. *p* < 0.05 was considered significant.

**Figure 3 pathogens-12-00794-f003:**
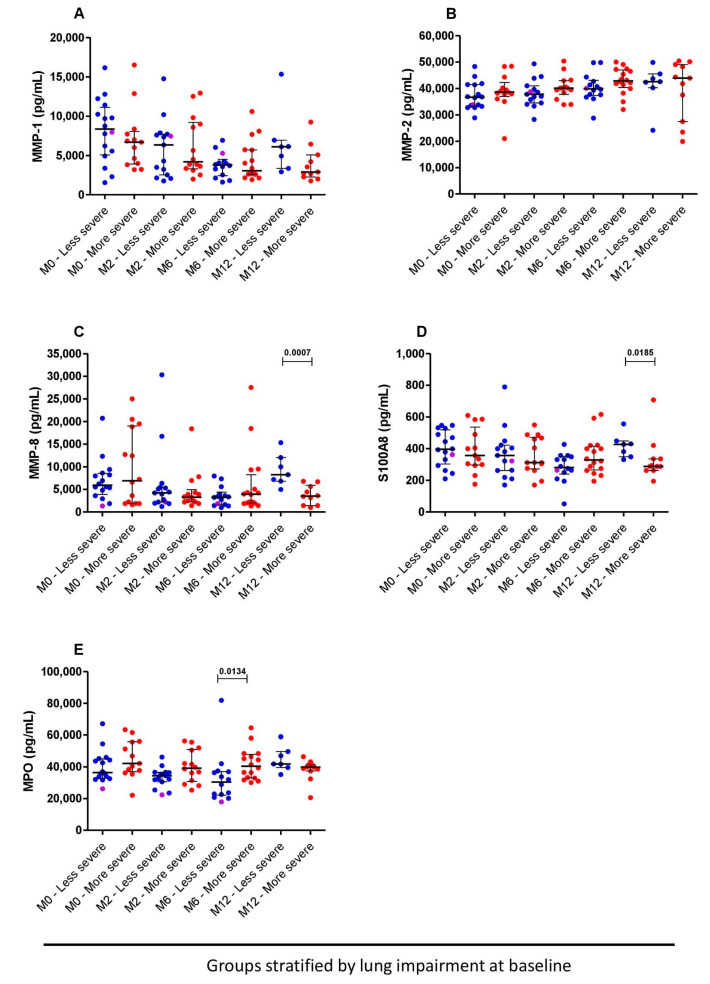
Comparisons of plasmatic biomarkers based on spirometry results from before TB treatment or on day 14: MMP-1 (**A**), MMP-2 (**B**), MMP-8 (**C**), S100A8 (**D**) and MPO (**E**) at baseline (n = 29), month 2 (n = 29), month 6 (n = 30) and month 12 (n = 17) between less (n = 16, blue dots) and more severe (n = 18, red dots) lung impairment. The more and less severe lung-impaired patients are represented by red and blue circles, respectively. The magenta dots represent the subjects with drug-resistant TB. Bars represent the median and interquartile range. Mann–Whitney tests were used for comparisons between the groups at different time points. *p* < 0.05 was considered significant.

**Figure 4 pathogens-12-00794-f004:**
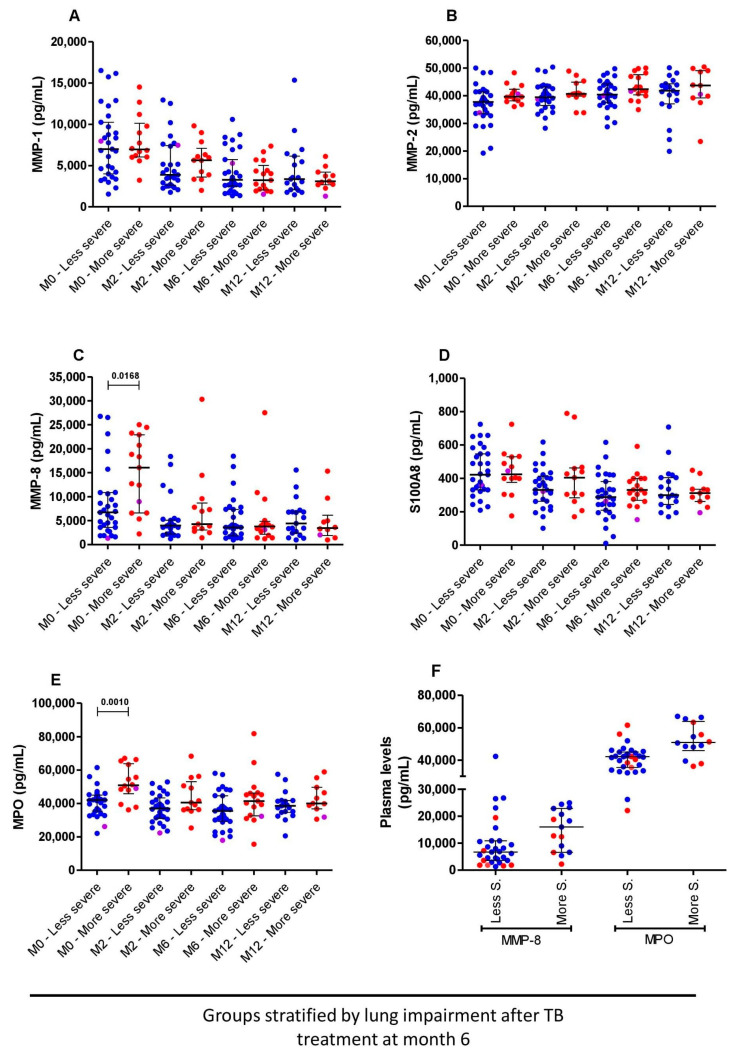
Comparisons of plasmatic biomarkers based on spirometry results of month 6: MMP-1 (**A**), MMP-2 (**B**), MMP-8 (**C**), S100A8 (**D**) and MPO (**E**) at baseline (n = 45), month 2 (n = 42), month 6 (n = 48) and month 12 (n = 31) between patients with more severe (n = 18) and less severe (n = 33) lung impairment. The more and less severe lung-impaired patients are represented by red and blue circles, respectively. The magenta dots represent the subjects with drug-resistant TB. (**F**) MMP-8 and MPO plasma levels at baseline of subjects with less and more severe lung impairment based on spirometry results at month 6, with red dots coding for subjects with more severe lung impairment at baseline. Bars represent the median and interquartile range. Mann–Whitney tests were used for comparisons between the groups at different time points. *p* < 0.05 was considered significant.

**Table 1 pathogens-12-00794-t001:** Characteristics of participants included in analyses.

	Total	PLHIV	HIV Negatives
*N*	68	47	21
Males, % (n/N)	57.35 (39/68)	53.19 (25/47)	66.67 (14/21)
Median age, years (min-max)	38.20 (19.0–60.6)	38.43 (23.8–60.6)	35.23 (19.0–59.2)
HIV- and ART ^a^ -naïve at BL ^b^, % (n/N)	48.94 (23/68)	48.94 (23/47)	NA
TB treatment, % (n/N)			
Standard	95.59 (65/68)	93.62 (44/47)	100 (21/21)
TB-DR ^c^	4.41 (3/68)	6.38 (3/47)	0 (0/21)
Smear result at BL, % (n/N)			
Negative	11.74 (8/68)	14.89 (7/47)	4.76 (1/21)
1+	8.82 (6/68)	4.26 (2/47)	19.05 (4/21)
2+	20.59 (14/68)	23.40 (11/47)	14.29 (3/21)
3+	36.76 (25/68)	36.17 (17/47)	38.09 (8/21)
Scanty	22.06 (15/68)	21.28 (10/47)	23.81 (5/21)
Ralph score, in median (IQR ^d^)			
At BL (n = 64)	15.00 (8.00–42.50)	12.0 (8.00–20.00)	20.00 (9.00–48.00)
At month 6 (n = 57)	5.00 (3.00–10.00)	5.00 (3.00–8.00)	7.00 (3.00–10.00)
Spirometry, % (n/N)			
Any lung impairment at BL	75.76 (25/33)	71.43 (15/21)	76.92 (10/13)
Any lung impairment at month 6	61.22 (30/49)	69.69 (23/33)	50.0 (9/18)
Any lung impairment at month 12	60.78 (31/51)	67.57 (25/37)	50.0 (8/16)
CD4 T-cell count, median in cells/mm^3^ (IQR) (n = 45)	NA	134.0 (66.50–343.00)	NA
Neutrophils, median in cells/mm^3^ (IQR)			
At BL (n = 67)	3.85 (2.78–5.33)	3.77 (2.67–5.24	4.15 (3.5–4.48)
At month 6 (n = 61)	1.61 (1.24–2.24)	1.73 (1.26–2.30)	1.53 (1.21–1.93)
At month 12 (n = 52)	1.79 (1.35–2.56)	1.63 (1.33–2.57)	2.14 (1.51–2.42)

^a^ ART: antiretroviral treatment; ^b^ BL: baseline; ^c^ TB-DR: TB-drug resistant; ^d^ IQR: interquartile range.

**Table 2 pathogens-12-00794-t002:** Correlation of the plasmatic biomarkers and neutrophil counts at baseline and months 6 and 12.

Time Point	MMP-1	Neutrophils at Baseline	Neutrophils at Month 6	Neutrophils at Month 12
Baseline	MMP-1	0.18 (*p* = 0.27)	−0.063 (*p* = 0.72)	−0.32 (*p* = 0.06)
	MMP-2	−0.42 (*p* = 0.0096)	−0.03 (*p* = 0.86)	0.07 (*p* = 0.69)
	MMP-8	−0.04 (*p* = 0.83)	0.12 (*p* = 0.48)	0.14 (*p* = 0.42)
	S100A8	0.19 (*p* = 0.23)	0.12 (*p* = 0.48)	−0.05 (*p* = 0.75)
	MPO	0.12 (*p* = 0.46)	0.09 (*p* = 0.46)	0.10 (*p* = 0.55)
Month 6	MMP-1	0.26 (*p* = 0.14)	−0.071 (*p* = 0.7)	−0.35 (*p* = 0.046)
	MMP-2	−0.42 (*p* = 0.014)	−0.22 (*p* = 0.22)	−0.05 (*p* = 0.79)
	MMP-8	0.22 (*p* = 0.22)	−0.02 (*p* = 0.89)	−0.01 (*p* = 0.95)
	S100A8	0.03 (*p* = 0.85)	−0.06 (*p* = 0.75)	−0.46 (*p* = 0.009)
	MPO	−0.0005 (*p* = 0.99)	−0.02 (*p* = 0.91)	−0.18 (*p* = 0.34)
Month 12	MMP-1	0.46 (*p* = 0.04)	−0.14 (*p* = 0.54)	−0.21 (*p* = 0.36)
	MMP-2	−0.28 (*p* = 0.22)	0.06 (*p* = 0.81)	−0.18 (*p* = 0.43)
	MMP-8	−0.22 (*p* = 0.16)	−0.36 (*p* = 0.11)	−0.08 (*p* = 0.71)
	S100A8	0.19 (*p* = 0.39)	−0.11 (*p* = 0.65)	−0.13 (*p* = 0.58)
	MPO	0.04 (*p* = 0.85)	−0.13 (*p* = 0.59)	−0.25 (*p* = 0.28)

**Table 3 pathogens-12-00794-t003:** Correlation of the plasmatic biomarkers and Ralph score at baseline and months 6 and 12.

Time Point	Biomarker	RS at Baseline	RS at Month 6
Baseline	MMP-1	−0.036 (*p* = 0.83)	0.31 (*p* = 0.08)
	MMP-2	−0.24 (*p* = 0.15)	−0.1 (*p* = 0.58)
	MMP-8	0.06 (*p* = 0.73)	0.18 (*p* = 0.32)
	S100A8	0.05 (*p* = 0.79)	0.22 (*p* = 0.22)
	MPO	0.07 (*p* = 0.68)	0.12 (*p* = 0.50)
Month 6	MMP-1	−0.13 (*p* = 0.46)	−0.01 (*p* = 0.96)
	MMP-2	−0.37 (*p* = 0.03)	−0.07 (*p* = 071)
	MMP-8	0.36 (*p* = 0.04)	0.08 (*p* = 0.67)
	S100A8	0.03 (*p* = 0.89)	−0.01 (*p* = 0.95)
	MPO	0.39 (*p* = 0.02)	0.10 (*p* = 0.58)
Month 12	MMP-1	−0.32 (*p* = 0.15)	0.10 (*p* = 0.67)
	MMP-2	−0.062 (*p* = 0.79)	0.23 (*p* = 0.33)
	MMP-8	0.07 (*p* = 0.77)	0.52 (*p* = 0.02)
	S100A8	0.08 (*p* = 0.72)	0.28 (*p* = 0.24)
	MPO	0.09 (*p* = 0.68)	0.35 (*p* = 0.13)

**Table 4 pathogens-12-00794-t004:** Spearman correlation among the plasmatic biomarkers at baseline and months 6 and 12.

Time Point	Biomarker	MMP-1	MMP-2	MMP-8	MPO
Baseline (n = 38)	MMP-1	NA	−0.24 (*p* = 0.15)		
	MMP-8	0.14 (*p* = 0.41)	0.09 (*p* = 0.57)		0.27 (*p* = 0.09)
	S100A8	0.63 (*p* < 0.0001)	0.01 (*p* = 0.097)	0.33 (*p* = 0.04)	0.06 (*p* = 0.74)
	MPO	0.01 (*p* = 0.94)	0.39 (*p* = 0.02)		NA
Month 6 (n = 37)	MMP-1	NA	−0.14 (*p* = 0.39)		
	MMP-8	0.07 (*p* = 0.66)	−0.04 (*p* = 0.82)		0.71 (*p* < 0.0001)
	S100A8	0.74 (*p* < 0.0001)	−0.06 (*p* = 0.73)	0.36 (*p* = 0.03)	0.38 (*p* = 0.02)
	MPO	0.11 (*p* = 0.51)	0.23 (*p* = 0.17)		NA
Month 12 (n = 42)	MMP-1	NA	−0.24 (*p* = 0.13)		
	MMP-8	0.14 (*p* = 0.37)	0.23 (*p* = 0.15)		0.54 (*p* = 0.0003)
	S100A8	0.42 (*p* = 0.0057)	0.28 (*p* = 0.07)	0.61 (*p* < 0.0001)	0.69 (*p* < 0.0001)
	MPO	0.19 (*p* = 0.22)	0.31 (*p* = 0.04)		NA

## Data Availability

Not applicable.

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
