# Peer review of "Effect of TB Treatment on Neutrophil-Derived Soluble Inflammatory Mediators in TB Patients with and without HIV Coinfection"

_pathogens, 2023, doi:10.3390/pathogens12060794_

Round 1

Reviewer 1 Report

The paper focused on evaluating the immune response as a marker of therapeutic management of tuberculosis in both immunocompetent and immunocompromised subjects. The topic has been the subject of general interest in all forms of lung diseases, from cancer to COPD and in this work on tuberculosis. In addition, it is known that many studies have focused on pulmonary cavitary lesions looking for markers of improvement of lung performance and response to therapy with a blood sample.

The costs of this investigation limit the use in some treatment centres.

The need arises not only for the diagnostic difficulties in some risk groups but also to evaluate the improvement and response of anti-tuberculosis treatment.

Compared to the works reported in the literature, the authors have enriched the data with spirometry.

 Other studies have examined the same markers during lung impairment from SARS-CoV-2 infection. The authors talk about it in the discussion.

I would invite you to stress about spirometry results that are not commented on in the discussion session. 

While table 1 I find incomplete because CD4 values are missing in immunocompetent; in fact, it is known that CD4 values are low even in HIV-negative subjects. In addition, I would add the CD4/CD8 ratio that better reflects the performance of the immune response of the HIV-positive subject.

I would only reduce Table 1 to patient demographics and remove the statistic by inserting it into the text since no significance was found. In contrast, the supplementary table could be inserted into the text. This table would give a lot of value as a visual impact even to those who are not experts in the field of research. 

Moreover, I recommend inserting the following references and commenting on them. 

V., P., do Valle, V. B., Fuzo, C. A., Fernandes, T. M., Toro, D. M., C., T. F., Basile, P. A., de Carvalho, J. C., Pimentel, V. E., Pérez, M. M., Oliveira, C. N., Rodrigues, L. C., Bastos, V. A., Tella, S. O., Martins, R. B., Degiovani, A. M., Ostini, F. M., Feitosa, M. R., Parra, R. S., . . . Sorgi, C. A. Matrix Metalloproteinases on Severe COVID-19 Lung Disease Pathogenesis: Cooperative Actions of MMP-8/MMP-2 Axis on Immune Response through HLA-G Shedding and Oxidative Stress. Biomolecules, 12(5), 604. https://doi.org/10.3390/biom12050604

Railwah, C., Lora, A., Zahid, K., Goldenberg, H., Campos, M., Wyman, A., Jundi, B., Ploszaj, M., Rivas, M., Dabo, A., Majka, S. M., Foronjy, R., El Gazzar, M., & Geraghty, P. Cigarette smoke induction of S100A9 contributes to chronic obstructive pulmonary disease. American Journal of Physiology-Lung Cellular and Molecular Physiologyhttps://doi.org/L-00207-2020

La Manna, M. P., Orlando, V., Paraboschi, E. M., Tamburini, B., Di Carlo, P., Cascio, A., Asselta, R., Dieli, F., & Caccamo, N. (2019). Mycobacterium tuberculosis Drives the Expansion of Low-Density Neutrophils Equipped With Regulatory Activities. Frontiers in Immunology, 10https://doi.org/10.3389/fimmu.2019.02761

Tiwari, D., & Martineau, A. R. (2023). Inflammation-mediated tissue damage in pulmonary tuberculosis and host-directed therapeutic strategies. Seminars in Immunology, 65, 101672. https://doi.org/10.1016/j.smim.2022.101672

Author Response

Responses

Question 1: I would invite you to stress about spirometry results that are not commented on in the discussion session. 

R: Thanks for your comment. We discussed the spirometry results at lines 344 – 357, and reinforced the involvement of neutrophils- related inflammatory biomarkers on lung tissues degradation. Additionally to that, we suggested to confirm our findings in a large sample size population and long-term follow up.

Question 2: While table 1 I find incomplete because CD4 values are missing in immunocompetent; in fact, it is known that CD4 values are low even in HIV-negative subjects. In addition, I would add the CD4/CD8 ratio that better reflects the performance of the immune response of the HIV-positive subject.

R: As described in “Material and methods – study participants section”, lines 86-87, we determined CD4 values only in those patients with positive result to HIV at the baseline visit. CD4 counts determination is recommended on WHO guideline for management of HIV infection. We agree that CD4/CD8 ratio better reflects the status of immune response in PLHIV but, unfortunately this test was not part of our study protocol and is not a routine for laboratory management of PLHIV.

Question 3. I would only reduce Table 1 to patient demographics and remove the statistic by inserting it into the text since no significance was found. In contrast, the supplementary table could be inserted into the text. This table would give a lot of value as a visual impact even to those who are not experts in the field of research.

R: Thanks for your suggestion. We agree, revised, and on table 1, the blood levels of C reactive protein and p value were removed. We kept other clinical and laboratory variables because we consider it relevant to describe our study population taking into account our study objective.

We also agree and inserted the supplementary tables into results section of the main text manuscript.

Question 4: Moreover, I recommend inserting the following references and commenting on

V., P., do Valle, V. B., Fuzo, C. A., Fernandes, T. M., Toro, D. M., C., T. F., Basile, P. A., de Carvalho, J. C., Pimentel, V. E., Pérez, M. M., Oliveira, C. N., Rodrigues, L. C., Bastos, V. A., Tella, S. O., Martins, R. B., Degiovani, A. M., Ostini, F. M., Feitosa, M. R., Parra, R. S., . . . Sorgi, C. A. Matrix Metalloproteinases on Severe COVID-19 Lung Disease Pathogenesis: Cooperative Actions of MMP-8/MMP-2 Axis on Immune Response through HLA-G Shedding and Oxidative Stress. Biomolecules12(5), 604. https://doi.org/10.3390/biom12050604

Railwah, C., Lora, A., Zahid, K., Goldenberg, H., Campos, M., Wyman, A., Jundi, B., Ploszaj, M., Rivas, M., Dabo, A., Majka, S. M., Foronjy, R., El Gazzar, M., & Geraghty, P. Cigarette smoke induction of S100A9 contributes to chronic obstructive pulmonary disse. American Journal of Physiology-Lung Cellular and Molecular Physiology. https://doi.org/L-00207-2020

La Manna, M. P., Orlando, V., Paraboschi, E. M., Tamburini, B., Di Carlo, P., Cascio, A., Asselta, R., Dieli, F., & Caccamo, N. (2019). Mycobacterium tuberculosis Drives the Expansion of Low-Density Neutrophils Equipped With Regulatory Activities. Frontiers in Immunology10. https://doi.org/10.3389/fimmu.2019.02761

Tiwari, D., & Martineau, A. R. (2023). Inflammation-mediated tissue damage in pulmonary tuberculosis and host-directed therapeutic strategies. Seminars in Immunology65, 101672. https://doi.org/10.1016/j.smim.2022.101672

R: We agree with your recommendation, inserted and commented. Please see the references 34, 50 and 51. Do Valle et al (2022) was included before in our discussion and references list.

Reviewer 2 Report

TB patients are treated with the combination of drugs for a period of 6 months (drug susceptible TB) to 2 years (drug resistant TB). Due to prolonged treatment duration, regular monitoring of TB patients is the key to the success of the treatment. Microscopy, culture, and molecular diagnosis on sputum samples are the common tests used for the diagnosis of active TB. Sputum test in TB patients is challenging for 2 reasons 1, HIV infection that reduces sensitivity and specificity of the tests and 2, low sputum production by TB patients after initiation of TB treatment. Therefore, identifying the new markers that associate with various stages (mild, moderate, and severe TB) of active TB after the initiation of TB treatment, would help in predicting the success of treatment. Active TB disease is characterized by neutrophilic inflammation, and this will subside once patients start taking antibiotics. Therefore, the level of neutrophils secreted inflammatory proteins will be changing fast over the course of TB treatment and hence can be used as a surrogate marker along with other established tests and markers for the monitoring of TB treatment. In this study, levels of MMPs, S100A8, MPO have been determined before and after initiation of TB treatment. The study includes both HIV positive with or without ART and HIV negative TB patients. TB incidence and associated morbidity and mortality increases manifolds in people living with HIV. Finally, data concludes that inflammatory proteins (MMP-1, MMP-8, S100A8 and MPO) which were elevated prior TB treatment, decrease over the course of TB treatment. This trend is noticed in both TB mono-infection and in HIV-TB co-infection. However, TB patients living with HIV, especially those not receiving ART treatments have elevated MMP-8 than the other groups in mid or after completion of TB treatment. The study represents the significant advancement in our quest of finding suitable soluble markers that can predict the outcome of TB treatment.

Major comment

Many graphs in figures 3 and 4 are inconclusive for the change in inflammatory mediators between the groups over the course of TB treatment. It is not clear why patients have been grouped into less severe (normal or minor lung impairments) and more severe (moderate and severe lung impairments) TB. All the inflammatory mediators (except MMP8 and S100A8 at 12 months) are insignificant between the groups, and it could be due to the incorrect grouping of the patients. Under this strategy, TB patients between the groups have less divergence for the lung impairments and that probably impacted the significance. The data may have definite pattern with significance for inflammatory mediators among the TB patients at different time points after initiation of TB treatment if patients are grouped into minor (normal lung), moderate (minor or moderate lung impairment) and severe (severe lung impairment) TB. Therefore, the author may consider plotting the data by grouping the patients in 3 groups as I mentioned above.

Minor comments

Introduction

Line 39, comma is not needed after both (HIV infection reduces sensitivity of both, the Gene Xpert molecular test).

Line 50, it appears, it is pulmonary treatment outcomes instead of pulmonary and treatment outcomes.

Line 57, it should be those levels or the levels of, instead of that levels of pro-inflammatory ………

Results

Figure 1, lines 163-164, hyphen is unnecessary in re-gardless, therefore, it should be regardless.

Figure 2, lines 297-298, Please give a clear conclusion instead of saying; in summary, these data show that several of the neutrophile derived inflammatory markers are elevated in PLHIV before and after TB treatment initiation.

Figure 3, lines 239-240, title lacks clarity, it can be written as …….at 6 months after initiation of Mtb treatment instead of … at 6 months.

English writing is satisfactory. I noticed errors with the use of hyphens and commas, and I mentioned them under the minor comments.

Author Response

Question 1:

Major comment: Many graphs in figures 3 and 4 are inconclusive for the change in inflammatory mediators between the groups over the course of TB treatment. It is not clear why patients have been grouped into less severe (normal or minor lung impairments) and more severe (moderate and severe lung impairments) TB. All the inflammatory mediators (except MMP8 and S100A8 at 12 months) are insignificant between the groups, and it could be due to the incorrect grouping of the patients. Under this strategy, TB patients between the groups have less divergence for the lung impairments and that probably impacted the significance. The data may have definite pattern with significance for inflammatory mediators among the TB patients at different time points after initiation of TB treatment if patients are grouped into minor (normal lung), moderate (minor or moderate lung impairment) and severe (severe lung impairment) TB. Therefore, the author may consider plotting the data by grouping the patients in 3 groups as I mentioned above.

R: Thanks for your comment and as described in the “material and methods – assessment of lung function and damage” section, the pulmonary function impairment was categorized as mild for FVC and FEV1/FVC-ratio >85% of predicted, moderate: FVC or FEV1/FVC-ratio 55-85% of predicted, severe: FVC or FEV1/FVC-ratio <55% predicted. However, due to the range interval of  FVC or FEV1/FVC ratio in moderate category, we assumed that would be relevant and robust if grouped moderate and severe as more severe impairment, and normal and mild as less severe impairment.

Nonetheless, the second reason was the sample size of patients with an available spirometry result at baseline/D14 and month 6 visits, as presented in the table below:

Normal

Mild

Moderate

Severe

Baseline/ D14

(n=34)

9

7

13

5

Month 6

(n=51)

19

14

10

8

However, although the sample size was a study limitation, our results show that for those started TB treatment as less severe lung impaired (normal or mild LI), have significantly higher plasma levels of MMP-8 and S100A8 at month 12 post-TB treatment initiation; and those started TB treatment as more severe LI (moderate or severe LI), the concentration levels of MPO was higher than those with less severe LI. Furthermore, patients with more severe LI at end of TB treatment, had higher levels of MMP-8 and MPO at baseline.

These three neutrophils-based inflammatory biomarkers have been involved in lung pathology and in our study, these may contribute for lung damage after TB treatment. Our study was exploratory and our results need to be confirmed in a large sample size study and assuming different grouping categorization of spirometry results.

Minor comments

Question 2: Introduction

Line 39, comma is not needed after both (HIV infection reduces sensitivity of both, the Gene Xpert molecular test).

Line 50, it appears, it is pulmonary treatment outcomes instead of pulmonary and treatment outcomes.

Line 57, it should be those levels or the levels of, instead of that levels of pro-inflammatory ………

Results

Figure 1, lines 163-164, hyphen is unnecessary in re-gardless, therefore, it should be regardless.

Figure 2, lines 297-298, Please give a clear conclusion instead of saying; in summary, these data show that several of the neutrophile derived inflammatory markers are elevated in PLHIV before and after TB treatment initiation.

Figure 3, lines 239-240, title lacks clarity, it can be written as …….at 6 months after initiation of Mtb treatment instead of … at 6 months.

R: We agree and revised it accordingly.

The hyphen is part of  journal manuscript template that cut a long phrase to continue in the following line.